# Simultaneously Embedding Indomethacin and Electrodeposition of Polypyrrole on Various CoCr Alloys from Ionic Liquids

**DOI:** 10.3390/ma15134714

**Published:** 2022-07-05

**Authors:** Florentina Golgovici, Florentina Gina Ionascu, Mariana Prodana, Ioana Demetrescu

**Affiliations:** 1Department of General Chemistry, Faculty of Chemical Engineering and Biotechnology, University Politehnica of Bucharest, 060042 Bucharest, Romania; florentina.golgovici@upb.ro (F.G.); flori.ionascu@yahoo.com (F.G.I.); ioana.demetrescu@upb.ro (I.D.); 2Academy of Romanian Scientists, 3 Ilfov, 050094 Bucharest, Romania

**Keywords:** CoCr alloys, polypyrrole, electropolymerization, indomethacin, drug release

## Abstract

The aim of the present investigation is the electrochemical deposition of polypyrrole films from choline chloride-based ionic liquids at various potential, period times and simultaneously an indomethacin embedding and release. The electrodeposition films were performed on CoCr commercial type Wirobond C (WBC) and, Heraenium CE (Hera) using as electroprocedures for deposition cyclic voltammetry and chronoamperometry. The morphology of obtained films was investigated using scanning electron microscopy (SEM). An FT-IR investigation of CoCr alloys before and after electrodeposition was able to identify the presence of polymer and drug. The research included an evaluation of the hydrophilic character of all studied samples and their electrochemical characterization in Tanni Zuchi artificial saliva. In the electrochemical study, the following methods have been used: open circuit potential, electrochemical impedance spectroscopy and potentiodynamic polarization. Indomethacin release from the polymeric film was determined using UV-VIS spectra. Based on Fick’s law of diffusion and indomethacin release profile, a kinetic law for release was established and discussed.

## 1. Introduction

Due to their remarkable mechanical and electrochemical stability in bioliquids, various CoCr alloys [1,2,3] are being investigated considering EU Regulatory framework, CMR Classification, and Toxicological Risks [4,5]. With such limitations, new trends have emerged that use well-known coating procedures enhancing the above alloy performances [6,7,8]. However, more procedures were combined in the last decade to introduce new composition structures and coatings of CoCr based alloys including antibacterial inhibitors [9,10,11]. Combined procedures to obtain synergy in reducing corrosion, and increasing antibacterial effect and biocompatibility were used as an extended strategy to get better biomaterials starting the beginning of our century on Ti, Ti alloys, and CoCr alloys, and nanotechnology was an aggressive positive impulse [12,13,14]. Simultaneously embedding drugs on coatings was often performed [15,16]. The electrodeposition of a polymer coating [17,18] has been used with various technologies and conditions, including the use of an ionic liquid as electrolyte, which enhances the surface properties of substrate [19,20] and electrodeposition efficiency [21,22]. When ionic liquids are an active part of electrodeposition, the drug delivery system could be improved due to an increase in solubility and loading of drugs. Because they are thermally stable, nonvolatile, and easily recycled, ionic liquids are considered green solvents. They are associated with advantages when used as electrolytes in polymerization. An example of conductive polymer electropolymerization which has been largely extended with ionic liquids as electrolytes [23] is polypyrrole. This is widely used in biomedical applications [24,25], including drug release. More recently, a larger group of ionic liquids with low toxicity known as deep eutectic solvents [26] have been an important ingredient in the development of drug release [23,27] capabilities. It is important to remark that simultaneous electropolymerization and drug loading seems to be a stability factor compared to other drug encapsulation procedures since the drug could be incorporated not only on the surface. Drug spreading could determine a better monitoring control of time for release and of kinetic process. Of course, the composition, amount, structure, and morphology of a particular drug play a significant role in the kinetic release. Numerous techniques have been built to facilitate conventional drug delivery systems by providing several advantages such as a decrease in the amount of dose required, an increase in the duration of drug release, and increased control of toxicity bioavailability. Various antibiotics such as gentamicine, vancomicyne tobramicyne and other local and systemic drugs [16,28,29] have been combined in different kinetics for more efficient antibiotic and anti-inflammatory surfaces [30]. The development of an interdisciplinary approach based on the association of different fields such as material science, pharmaceutical science, computer technology, and various other technological systems [31,32] is a new strategy for producing devices that provide more economical therapies. The present work describes a novel approach applied to two CoCr commercial alloys. More specifically, in this approach, this paper introduces very complex coatings including simultaneously entrapment indomethacin and polypyrrole on CoCr alloys using as electroprocedures for deposition cyclic voltammetry and chronoaperometry performed in choline chloride-based ionic liquids. Based on Fick’s law of diffusion and indomethacin release profile, a kinetic law for release was established and discussed.

## 2. Materials and Methods

### 2.1. Materials

As metallic substrates, two types of commercial CoCr alloys commonly used in dental medicine were used, and their chemical composition is shown in Table 1. Wirobond C alloy (WBC) was supplied by Bego (Lincoln, RI, USA) and Hearenium CE alloy (Hera) was purchased from Heraeus (Hanau, Germany).

Choline chloride (Aldrich, St. Louis, MI, USA, 99%), acetonitrile (Aldrich, 99.8%), malonic acid (Aldrich, 99.8%), and indomethacin (Alfa Aesar, Haverhill, HA, USA, 99+%) were used exactly as received. Pyrrole (Py) (Aldrich, 98%) was distilled and stored under nitrogen until use. The idea of using ionic liquids as electrolyte for Py electropolymerization started from the fact that indomethacin is poorly soluble in water. Thus, an eutectic mixture of choline chloride-malonic acid (1:1 molar ratio) with additions (1:1 by volume) of acetonitrile (ACN) to increase conductivity was used as electrolyte. Choline chloride was combined with malonic acid and heated at 80 °C with gentle stirring until a homogeneous, clear liquid was formed. The as-prepared ionic liquid was cooled to room temperature and ACN (1:1 by volume) was added, followed by 0.5 M Py. Finally, 0.05 M Indo was added with continuous stirring.

### 2.2. PPy and PPy-Indo Electropolymerisation on CoCr Alloys

Electrochemical techniques, such as cyclic voltammetry and chronoamperometry, were used to get information on the electropolymerization of PPy and PPy-Indo coatings from ionic liquids-based electrolytes. For all electrochemical polymerization, an AutoLab 40 potentiostat/galvanostat (Radiometer Analytical SAS, Lyon, France) was used. As an electrochemical cell, a single-compartment glass cell was used, with CoCr alloy samples with an exposed constant geometrical surface of 1.57 cm^2^ acting as working electrodes. The counter electrode was a high surface area platinum (Pt) plate, and the quasi-reference electrode was a silver wire [16]. Before each experiment, the CoCr alloy working electrodes were mechanically polished using 1200-grit SiC abrasive paper. Finally, the samples were cleaned ultrasonically in acetone for 10 min to remove contaminants from the surface before being air-dried.

Following the formation of the polymer coating, the electrodes were thoroughly cleaned with distilled water and air-dried. All investigations were carried out in stationary circumstances at room temperature. The voltammetric measurements were taken at scan rates ranging from 10 to 50 mV s^−1^. The synthesis of PPy and PPy-Indo coatings was also studied under potentiostatic conditions at different applied potentials for a 600 s constant polarization period from various ionic liquids-based electrolytes containing the monomer (Py), with and without Indo.

### 2.3. Characterization of the Coatings

Scanning electronic microscope (SEM) (Hitachi 82830 SU EDS) was used to examine the surface morphology of the produced PPy and PPy-Indo coatings. Electron acceleration voltage was 10 kV.

Fourier transform infrared (FTIR) spectroscopy (Jasco FT/IR 4000, Tokyo, Japan) was employed to determine their chemical composition. At 30 acquisitions per sample, spectra were obtained from 4000 to 600 cm^−1^ with a resolution of 4 cm^−1^.

The surface wettability variables were evaluated by measuring the contact angle artificial saliva type Tani-Zuchi [33] with the following chemical composition: 1.5 g L^−1^ KCl, 1.5 g L^−1^ NaHCO_3_, 0.5 g L^−1^ NaH_2_PO_4_, 0.5 g L^−1^ KSCN, and 0.9 g L^−1^ lactic acid per 1 L of double-distilled water. A CAM 100 compact contact angle meter was employed, and the average value of five measurements taken on five different regions of the specimens was considered.

Using Tani-Zuchi artificial saliva as an electrolyte and the same three electrode cells, the protective properties of the produced PPy and PPy-Indo coatings were also evaluated. Potentiodynamic polarization curves were obtained at a sweeping rate of 5 mV s^−1^ against an Ag/AgCl reference electrode and a Pt mesh counter electrode. 

### 2.4. Drug Release Studies

The release of Indo molecules from polymeric coatings was investigated using optimized PPY-Indo coating specimens placed in a 20 cm^3^ artificial saliva. The polymer coatings were thoroughly cleaned with distilled water and dried before the drug release trials. The drug release curves were constructed using periodic withdrawals of the solution containing the drug releasing specimens (three aliquots of 100 µL each). The calibration curve for Indo was obtained using a Perkin-Elmer UV-Vis spectrometer (Perkin-Elmer, Shelton, CT, USA) at λ = 266 nm. Calibration standard solutions of Indo were prepared in the concentration range from 0 to 10 mg L^−1^ and Indo calibration straight line was achieved. Using the same analytical approach, the initial amount of indomethacin in the polymer coating was determined and the average of three measurements was used for data analysis.

## 3. Results

### 3.1. Electrochemical Preparation of PPy and PPy-Indo Coatings 

The first step was the study of PPy layers growing from ionic liquids as an electrolyte on the two types of CoCr alloys. Cyclic voltammetry was used to observe the behavior of CoCr-based alloys in an ionic liquid consisting of choline chloride-malonic acid- acetonitrile and 0.5 M monomer. Figure 1 shows examples of multicycle voltammograms (CVs) recorded during PPy development on WBC (Figure 1a) and Hera (Figure 1b) alloys. The red line shows the cyclovoltammograms obtained for the CoCr studied alloys in electrolyte without the Py monomer. 

As we can see from Figure 1, for CoCr studied alloy, we obtain a different form of the first cycle from the following recorded cycles. A flattened anodic peak can be observed in the region of 0.92–1.01 V/Ag on WBC alloy (Figure 1a), respectively at around 0.94–1.06 V/Ag on Hera alloy (Figure 1b), signaling the oxidation of the pyrrole monomer, or the start of polymer film growth. The anodic peaks that can be observed on further anodic scanning for both WBC and Hera alloys are attributed to the oxidation process of the Cl^−^ anions existing in the ionic liquid electrolyte. It can also be seen that, as the number of cycles increases, the value of the anodic peak current density associated with the electropolymerization of the pyrrole increase, so we can talk about a progressive increase in the electrical charge. If the electropolymerization process proceeds, the potential required to pyrrole to polymerize may become less positive than that necessary for monomer oxidation [34]. The corresponding reduction peaks can be observed during the cathodic scan. For these, too, the current density increases as the number of scan cycles increases. This can be associated with the expulsion of a fraction of chloride ion species from the polymer film because of its reduction from the oxidized to the neutral state [35]. We can see a relatively increased polymerization rate on the Hera type alloy, which is caused by higher current density values. This behavior could be explained by a tiny difference in the potential necessary to start Py oxidation on the two CoCr alloys, indicating the synthesis of various oligomeric Py species [34].

To be able to observe that the ionic liquid does not determine the appearance of major peaks in the potential range of interest from the pyrrole electropolymerization point of view, the cyclic voltammetry curves were recorded for the CoCr alloys studied in the ionic liquid that did not contain monomer. These are presented as inserts for each alloy studied. In both cases, a current plateau can be seen over a very wide range of potentials. On both the anodic and the cathodic branches, the values of current densities are very low compared to those recorded for the electrolyte containing pyrrole.

After electropolymerization, WBC and Hera alloys were completely coated by a black, homogeneous, and adherent PPy layer.

The next step was to obtain in potentiostatic conditions the polymer films on the surface of the CoCr alloys. Three potential values were used for electrosynthesis, and the time remained the same, namely 600 s.

The value of the charge is extremely important in electropolymerization because it allows us to estimate the thickness of the coating. In Figure 2, we presented the examples of the recorded charge-time plots for the studied CoCr alloys during potentiostatic polymerization from ionic liquid containing 0.5 M Py.

As can be seen, an applied potential increase from 1 to 1.2 V (vs. Ag quasireference) determines an increase in the charge values required for anodic polymerization [36]. Similar behavior was established for the second studied alloys, with slightly higher values for charge recorded for the Hera-type alloy. 

When 0.025 M indomethacin was introduced into the electrolyte, the behavior obtained is shown in Figure 3. This shows that, as the applied potential is increased, the value of the electric charge increases, suggesting the formation of a thicker conductive coating.

Comparing the charge values in Figure 3 with those shown in Figure 2, it can be said that the introduction of the drug leads to an increase in the polymerization charge in all studied cases. In this case too, it should be noted that the electrodeposition of the PPy-Indo coating on Hera type alloy is slightly faster than on WBC type CoCr alloy. From the analysis of Figure 2 and Figure 3, it can be said that a consistent rate of electropolymerization with no evidence of a decline in the conductivity of the deposited polymer was observed [37,38].

The optimum parameters for PPy and PPy-Indo growth on WBC and Hera type CoCr based alloys with a high degree of reproducibility were found to be obtained by the chronoamperometry at a value of the potential applied of 1.1 V (vs. Ag quasireference). These parameters have been established following numerous studies at different deposition periods and applied potentials associated with the quality of the prepared films, respectively.

For further studies, the PPy and Indo-doped PPy coatings electrosynthesized from choline chloride-based electrolytes on the two CoCr alloys were obtained at the same total charge value of 2 C.

### 3.2. Morphological and Structural Characterisation of Indo Doped PPy Coatings

#### 3.2.1. Scanning Electron Microscopy (SEM) Measurements

SEM microscopy was used to obtain information about the morphology of the PPy and PPy-Indo coatings simultaneously potentiostatic electrodeposited on the two types of commercial CoCr alloys. Figure 4 presents examples of the SEM micrographs of polymeric layers electrochemically synthesized from ionic liquid-based electrolytes on the two investigated metallic substrates. The micrographies show aggregates formed as micrometre-sized clusters.

The characteristic cauliflower-like structure of polypyrrole can be observed both in the coatings obtained on WBC and in those electrodeposited on the Hera alloy. For both CoCr studied alloy types, the surface morphology of the electrosynthetized PPy-Indo polymer films is quite granular. The deposition covered the entire surface of the alloys, the film is homogeneous, and no significant differences were observed between alloy samples covered with PPy (Figure 4a,c). By adding Indo into PPy, the dimensions of granules increased from 50–200 nm to 300–500 nm for both alloys (Figure 4b,d).

Furthermore, the SEM micrographs show that there are no insoluble precipitates of the drug molecules on the surface. This demonstrates the total incorporation of Indo into PPy film.

#### 3.2.2. FTIR Measurements

FTIR analysis has also been used to investigate the chemical structure of the electrodeposited coatings (Figure 5). As can be seen, in the FIIR spectra recorded for the case of polypyrrole-coated alloys, specific peaks are found. So, the presence of N–H stretching vibrations of the pyrrole ring [39,40] can be proved by the existence of a broad shoulder in the area 3400–3200 cm^−1^ shown in Figure 5a,b. The peak at 1566 cm^−1^ is primarily caused by the intercycle C–C stretching vibration in PPy, whereas the band centered at 1163 cm^−1^ can be associated with C–N stretch and the peak at 957 cm^−1^ is correlated with ring stretch from pyrrole [41,42].

The presence of indomethacin in the coatings in which the simultaneous embedding of the drug was made during the electropolymerization of pyrrole, is proved by the appearance of the main peaks of indomethacin in the FTIR spectra. The peak recorded at 2900 cm^−1^ corresponds to Ar–H stretching vibration in aromatics (Ar), the one at 2860 cm^−1^ can be associated with C–H stretch, and 1740 cm^−1^ peak is assigned to the C=O and C–O stretches. O–CH_3_ deformation and the presence of an aromatic ring of indomethacin can be observed through the peaks at 1468 cm^−1^, respectively 708 cm^−1^ [16,43,44]. 

In conclusion, it can be said that the presence of indomethacin in the polypyrrole structure has been demonstrated for the studied samples.

### 3.3. Contact Angle Evaluation

Since wetting phenomena influence cell adhesion and proliferation on CoCr-based alloys coated with polypyrrole, as well as on those coated with polypyrrole and drug, contact angle values were measured for uncoated CoCr-based alloys and those coated with polymer films obtained electrochemically from ionic liquids, as well as polymer films electrodeposited simultaneously with indomethacin. 

Contact angle analysis may provide useful information regarding the hydrophilicity and hydrophobicity of a material’s surface, which is typically dictated by the chemical composition and surface morphology. As shown from the literature [45,46], hydrophilic surfaces have contact angles between 0° and 90°, whereas hydrophobic surfaces have contact angles greater than 90°. 

The average values obtained from five measurements for each case are shown in Table 2. The tests were performed with Tani- Zuchi artificial saliva.

As we can see from Table 2, uncoated CoCr alloys exhibited significant hydrophilicity, which was more pronounced for the WBC-type alloy. The presence of the electropolymerized PPy coating reduced the contact angle values for WBC and Hera CoCr-based alloys at approximately 27° and 21°, respectively. This means that polymer films have an affinity for Tani-Zuchi artificial saliva. This is desirable because PPy is obtained from ionic liquids, and it has been observed that it can absorb an amount of liquid equal to 10% of its mass.

Furthermore, Indo doping contributed to a slight decrease in contact angle value, implying an enhancement in hydrophilic properties. This behavior could be attributed to the slightly porous shape of the Indo doped PPy coatings.

In conclusion, it can be said that the surfaces of CoCr alloys exhibit hydrophilic properties, and the electrodeposition of polymeric films with or without drugs has an impact on the surface wettability. This can be explained by the increase in hydrophilic drug additives.

Such phenomena are expected to increase the adhesion and proliferation of cells over the material coated with polypyrrole or polypyrrole with a drug (indomethacin).

### 3.4. Electrochemical Characterisation

The protective characteristics of the PPy and Indo doped PPy coatings on both WBC and Hera type CoCr alloy substrates have been assessed in Tani-Zuchi artificial saliva as electrolyte compared to uncoated alloys in the same electrolyte. 

#### 3.4.1. Open Circuit Potential Tests

For uncoated and coated with PPy and PPy-Indo CoCr alloys, open circuit potential (OCP) tests were taken for 600 s. Figure 6 shows how potential values rapidly shift towards electronegative values during the first seconds of immersion in artificial saliva. This is followed by a noticeably slower decrease toward electronegative levels, which culminates in a plateau.

It can be said that the PPy or PPy-Indo coating of the two CoCr-based alloys studied leads to a slightly more electronegative value of the stabilized potential compared with uncoated alloy.

#### 3.4.2. Electrochemical Impedance Spectroscopy Measurements

Since conductive biomaterials’ electrochemical characteristics are crucial for their function as bioelectrodes or tissue engineering scaffolds to efficiently mediate electrical impulses [47], the next electrochemical method used for the analysis was electrochemical impedance spectroscopy.

The spectra recorded at open circuit potential in Tani-Zuchi artificial saliva are presented as Nyquist and Bode diagrams in Figure 7 for WBC type alloy and Figure 8 for Hera type alloy.

As can be seen from the Nyquist diagram recorded for PPy or PPy-Indo coated WBC, shown in Figure 7a, in the case of the uncoated alloy a single capacitive semicircle appears, whereas for the coated alloy the appearance of two interfaces can be observed. The first capacitive semicircle is attributed to the polymer coating. It can be seen that doping the polypyrrole with the drug leads to an increase in the diameter of this capacitive semicircle, indicating better corrosion protection of this type of coating. Furthermore, all the PPy or PPy-Indo coated CoCr alloys had much lower impedance than uncoated CoCr-based alloy electrodes [48]. Correspondingly, in the Bode plot shown in Figure 7b, only one time constant is recorded for the uncoated alloy, while two-time constants, one at a medium-high frequency and the second one at a lower frequency, are presented for the coated alloy. The one at high frequencies is associated with the polymer coating or drug-doped polymer, the second one is associated with the electrical double layer created at the coating-electrolyte interface. A decrease of the maximum phase angle value from −77° in the case of uncoated alloy to −68°, respectively −60° in the case of PPy or PPy-Indo coated WBC can also be observed.

Similar behavior was recorded for the spectra recorded for the Hera-type alloy in Tani-Zuchi artificial saliva and shown in Figure 8a as Nyquist diagrams, showing the dependence of imaginary part of impedance vs. real part of the impedance and Figure 8b as Bode diagrams, pointing simultaneously two dependencies: impedance modulus vs. frequency and phase angle vs. frequency. The decrease in the maximum phase angle values from −74° (for uncoated Hera alloy) to −42° (for PPY-Indo coated Hera alloy) is more pronounced than in the case of WBC type CoCr based alloy.

The experimental impedance data were analyzed using ZView software (Scribner Associates Inc., Southern Pines, NC, USA) and the electrical equivalent circuit (EEC) as seen in Figure 9. Table 3 displays the values derived for the corresponding electrical circuit elements. When the experimental data were fitted, a value of Chi-quadrate (2) of about 10^−2^ was found, suggesting that the fitting errors were small.

The model provided in Figure 9a was applied for the uncoated CoCr alloys, which is a simple Randles circuit consisting of the electrolyte resistance between the working and reference electrodes (Rs) and a parallel combination of the double layer capacitance, CPE_dl_. To describe the electrolyte–substrate interface, a charge transfer resistance, R_ct_, was provided to the equivalent circuit. The model in Figure 9b was employed for the PPy or PPy doped with drug coated CoCr samples. Specifically, the proposed model for uncoated alloys was completed with the coating capacitance (CPE_coat_) and the coating resistance (R_coat_).

The obtained fitting parameters presented in Table 3 reveal high values of the coating resistance R_coat_, and low values of capacitance component of CPE, values that prove the high capacitive behavior of the coatings. The CPE-P independent parameter presents values close to 1, evincing the ideal capacitors for the studied samples. The highest Rf value was obtained for the PPy-Indo coating, indicating the best anti-corrosion protection in Tani-Zuchi artificial saliva.

#### 3.4.3. Potentiodynamic Tests

In order to apply another electrochemical characterization method, the polarization curves for the coated and uncoated CoCr based alloys were recorded. As can be seen from Figure 10, coating the CoCr alloys with PPy or PPy-Indo leads to a shift towards more electropositive values of the corrosion potential. 

Also, a decrease in the corrosion current density value when the metal is coated can be observed for both the WBC alloy (Figure 10a) and the Hera alloy (Figure 10b).

Corrosion kinetic parameters were calculated from the polarization curves illustrated in Figure 10a,b using two methods: the Tafel slope extrapolation method and polarization resistance method. Table 4 shows the values obtained for some kinetic corrosion parameters, such as corrosion potential (E_corr_), corrosion current density (i_corr_), gravimetric index (kg), corrosion rate (CR), anodic slope (b_a_), cathodic slope (b_c_), and polarization resistance (R_P_). The corrosion current density values obtained by the two approaches were comparable. The polarization resistance is noted as Rp and was calculated from the Stern–Geary equation [49]. This parameter is representative for the degree of protection at the surface. The higher the value obtained for the polarization resistance, the lower the value of the corrosion rate value will be. Both corrosion current density (i_corr_) and corrosion rate (CR) were determined according to the ASTM-G59 standard [50]. The polarization resistance of the investigated PPy or PPy-Indo CoCr coated alloy increases compared to uncoated alloy, showing a decrease of the corrosion rate of each of the studied biomaterials in Tani-Zuchi artificial saliva. All parameters were statistically analyzed, and the results are reported as mean ±1 standard deviation.

In conclusion, it can be said that the coating of CoCr base alloy with PPy-Indo layers exhibits better corrosion protection as compared to the PPy ones, for both studied alloy substrates. This agrees with the results obtained by electrochemical impedance spectroscopy. Both types of coating give both WBC type and Hera type CoCr alloys corrosion protection more related to the inhibition of the anodic reaction. Comparing the behavior of the two alloys in Tani-Zuchi type artificial saliva, it can be said that for the Hera type alloy higher values for corrosion rate were recorded than for the WBC type alloy and consequently lower values for polarization resistance.

### 3.5. Indomethacin Release from the Polypyrrole Coating

The drug-loaded specimens were immersed in Tani-Zuchi artificial saliva at room temperature to determine the release of indomethacin from the polymer films. 

Initially, the calibration curve was performed at the highest absorption band, 266 nm, to determine the linear regression. Thus, the line equation was y = 56.4x + 0.00616 and the correlation coefficient obtained was r^2^ = 0.9987, indicating a significant linear regression (*p* < 0.0001) and adequate linearity for concentrations up to about 10 µg/mL. 

Figure 11 illustrates the indomethacin release profiles from doped conducting polymer coatings electrosynthesized on WBC and Hera CoCr based alloys using a chlorine chloride-based electrolyte containing 0.5 M Py and 0.02 5M Indo.

The cumulative drug release, designated R, is the fraction (percentage) of the active species released at time t in Tani-Zuchi artificial saliva, determined with Equation (1), as: (1)$$R=\frac{q}{q_{0}}\times 100$$
where q_0_ is the initial amount of Indo in the coating and q is the amount of drug released at time t.

As we can see from Figure 11a, for PPy-Indo coatings on WBC alloy, an initial burst release effect up to about 27% during the first 10 h may be noticed, with half the amount of drug incorporated in the polymer coating being released in about 40 h, followed by a prolonged release phase for the next 180 h and then a relatively constant release profile up to 360 h.

In the case of the drug embedded in the electrosynthesized choline chloride-based electrolyte coating on the Hera-type alloy, the release is shown in Figure 11b, pointing to an initial burst release of about 17% during the first 10 h, followed prolonged release phase up to 250 h and a continuous prolonged release phase up to 460 h.

As previously said, PPY-Indo electrochemically generated coatings may be able to deliver indomethacin for longer periods. The cumulative indomethacin released fraction after 360 h for WBC alloy used as a substrate is in the region of 98.9–99.8%, and after 460 h for Hera alloy used as a substrate is in the range of 97.97–99.3%.

In the case of the doped conducting polymer coatings electrosynthesized on WBC, this trend predicts a quicker drug release. In general, the morphological properties of the films may affect the diffusion of active compounds towards solution [51].

Finally, different mathematical models were applied to indomethacin release from the PPy coatings data such as Korsmeyer-Peppas, Higuchi, Hixson-Crowell, Baker-Lonsdale, Zero Order, and First Order). However, the mathematical model described by Korsmeyer-Peppas (Equation (2)) had the best fit for the experimental release curves (r^2^).

The Korsmeyer–Peppas model, widely known as the power law model, is based on a simple semi-empirical equation that is frequently used in pharmaceutical research to characterize release kinetics [52,53]. This equation is expressed as:(2)$$R={\mathrm{kt}}^{n}$$
where R is the cumulative drug release fraction at time t, the diffusional exponent characterizing the mechanism of the drug release is noted by n and the parameter which incorporate structural and geometric characteristics of the drug/polymer system is the kinetic constant, k. The Korsmeyer–Peppas model may be seen as a decision parameter to identify drug transport mechanisms where n is used to differentiate between the various drug release mechanisms.

The computed parameters for Indo doped PPy coatings electrochemically produced from choline chloride-based electrolytes on CoCr alloy substrates are shown in Table 5.

The release coefficient was n > 0.5 for Indo from PPy coatings electrodeposited on CoCr alloys suggesting a non-Fickian behavior which also takes into consideration the erosion of the polymer chain [53,54]. 

The value close to 1 for the correlation coefficients (r^2^) determined during the fit of the Korsmeyer–Peppas model to the experimental release data for all PPy-Indo coatings suggests that the indomethacin release can be described using this model. 

The higher value for the kinetic constant obtained when WBC was used as substrate suggests a fast release of the drug from the PPy coatings. This is more pronounced compared to the Hera alloy used as a substrate for simultaneously embedding indomethacin in the polymeric layer.

## 4. Conclusions

Simultaneous indomethacin embedding during polypyrrole electrosynthesis from choline chloride-based ionic liquids has been made for the first time on various CoCr commercial alloys. Compared to aqueous electrolytes, this type of electrolyte allows a greater solubility for indomethacin.

SEM micrographs of the PPy and PPy-Indo coatings revealed a granular surface morphology and no drug-molecule insoluble precipitates. By adding Indo into PPy, the dimension of clusters increases from 200 nm to 500 nm, the surface of the alloys being uniformly covered by the PPy-Indo film. The presence of indomethacin in the polypyrrole structure has been confirmed by recorded FTIR spectra. 

The enhancement of hydrophilic characteristics when the CoCr alloys studied were coated with PPy or PPy-Indo was determined by measuring the contact angle.

Also, coating the commercial CoCr alloys studied with polymer films or polymer films with incorporated drugs led to improved anticorrosive properties of the samples in Tani-Zuchi artificial saliva.

Indomethacin release tests showed that the PPy-Indo coatings may deliver the drug molecule for longer periods of 460 h in the Hera alloy substrate case. For the polymeric coating doped with drug electrosynthesized on WBC alloy, half the amount of drug incorporated in the polymer coating is released in about 40 h. The maximum amount released was determined to be around 99.8%, indicating that the polypyrrole layers worked as an effective reservoir for indomethacin. A non-Fickian behavior was established as a mechanism for the release profiles of the indomethacin out of the polymer layer.

## Figures and Tables

**Figure 1 materials-15-04714-f001:**
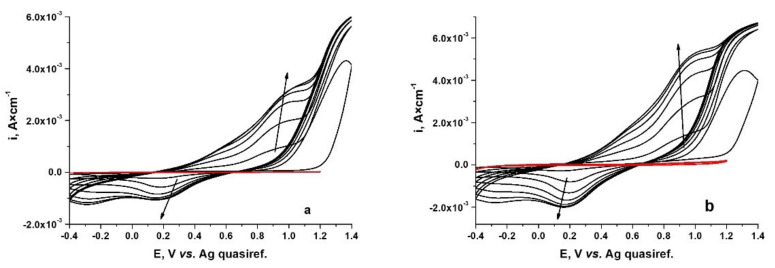
Cyclic voltammograms on WBC (**a**) and Hera (**b**) CoCr alloys during electrosynthesis of PPy from ionic liquids electrolyte containing 0.5 M Py. The scan rate was 10 mV s^−1^. (Red line: CVs for CoCr alloys in electrolyte without monomer).

**Figure 2 materials-15-04714-f002:**
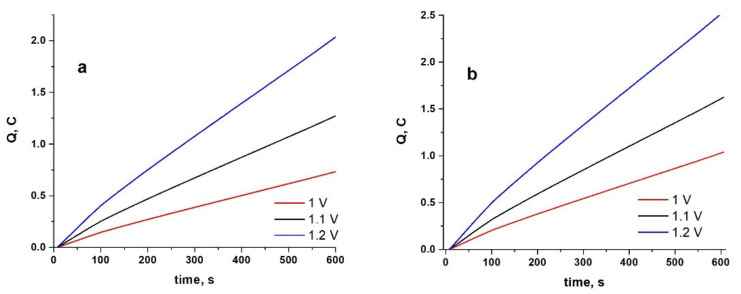
Charge-time plots during electrosynthesis of PPy at different constant potentials for 600 s on WBC (**a**) and Hera (**b**) CoCr alloys from ionic liquid electrolyte containing 0.5 M Py.

**Figure 3 materials-15-04714-f003:**
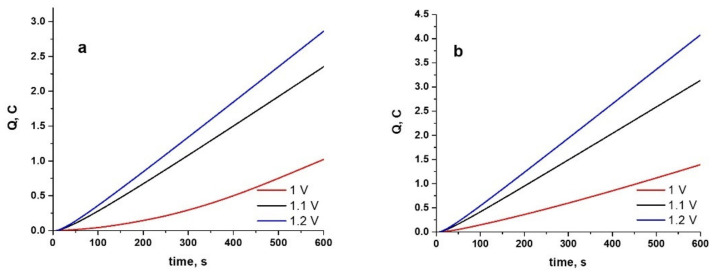
Charge–time plots during electrosynthesis of PPy at different constant potentials for 600 s on WBC (**a**) and Hera (**b**) CoCr alloys from ionic liquid electrolyte containing 0.5 M Py and 0.025 M Indo.

**Figure 4 materials-15-04714-f004:**
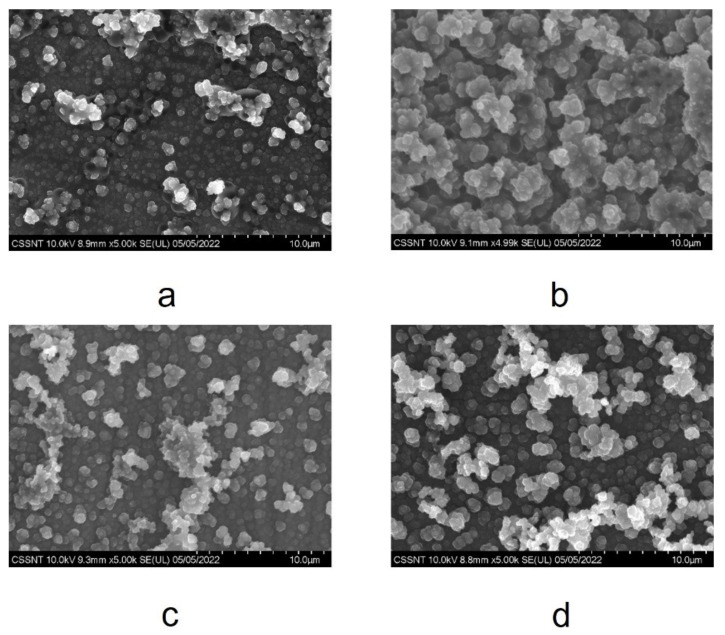
SEM micrographs at the same magnifications (5 kx) for: (**a**) PPy on WBC; (**b**) PPy-Indo on WBC; (**c**) PPy on Hera and (**d**) PPy-Indo on Hera types of alloys.

**Figure 5 materials-15-04714-f005:**
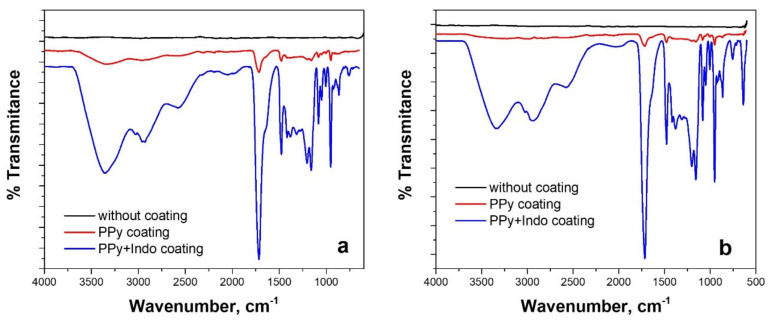
FTIR spectra of electrochemically synthesized PPy and PPy-Indo coatings on WBC (**a**) and Hera (**b**) CoCr-based alloy substrates from ionic liquid-based electrolyte.

**Figure 6 materials-15-04714-f006:**
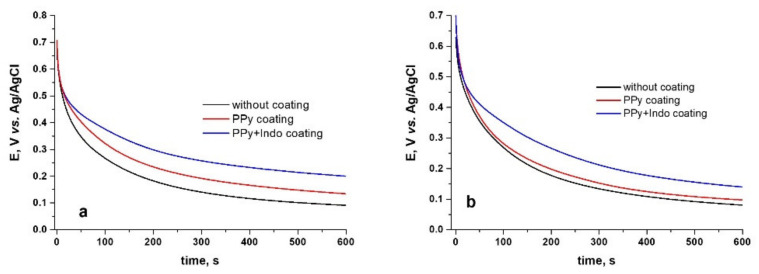
Open circuit potential behavior in Tani-Zuchi artificial saliva at 37 °C for the coated and uncoated WBC (**a**) and Hera (**b**) CoCr-based alloy.

**Figure 7 materials-15-04714-f007:**
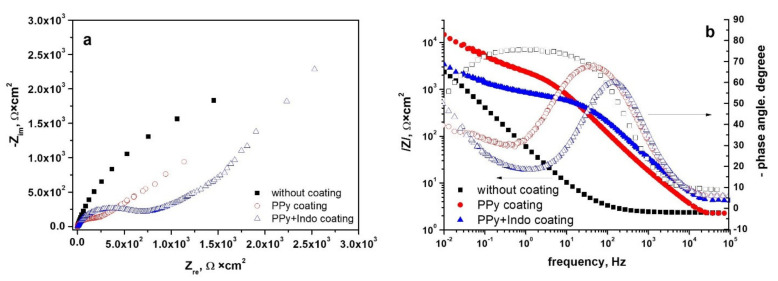
Nyquist (**a**) and Bode (**b**) diagrams for uncoated and PPy or PPy-Indo coated WBC type CoCr based alloys in Tani-Zuchi artificial saliva.

**Figure 8 materials-15-04714-f008:**
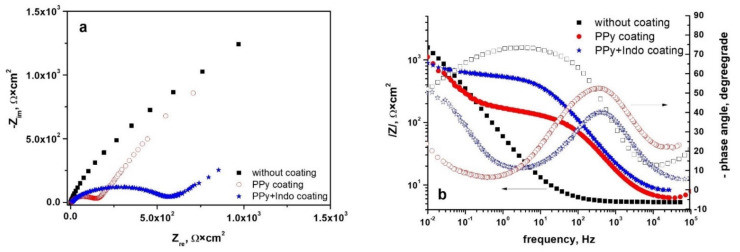
Nyquist (**a**) and Bode (**b**) diagrams for uncoated and PPy or PPy-Indo coated Hera type CoCr based alloys in Tani-Zuchi artificial saliva.

**Figure 9 materials-15-04714-f009:**
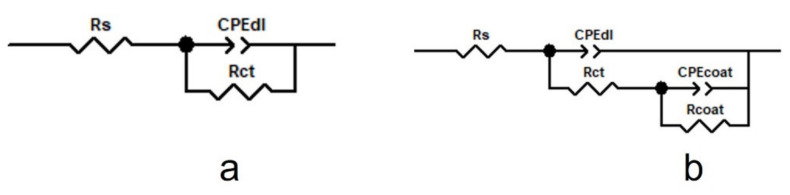
Models for the equivalent electrical circuit proposed for fitting the experimental impedance spectra, (**a**) for uncoated alloy and (**b**) PPy or PPY-Indo coated alloy.

**Figure 10 materials-15-04714-f010:**
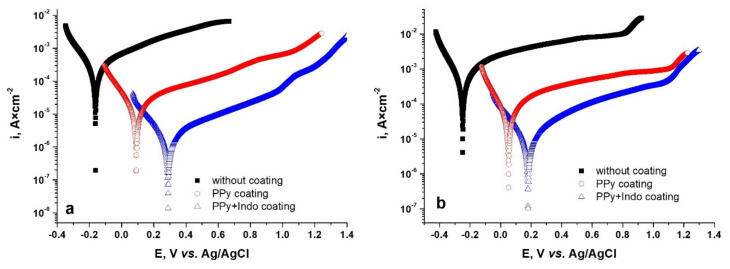
Polarization curves for uncoated and PPy or PPy-Indo coated WBC (**a**) and Hera (**b**) CoCr based alloys in Tani-Zuchi artificial saliva.

**Figure 11 materials-15-04714-f011:**
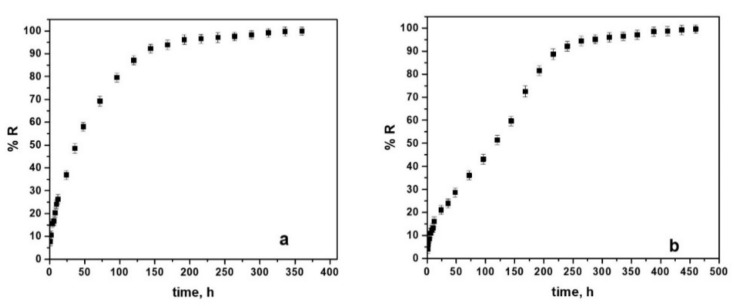
Indomethacin release profiles from the PPy-Indo coatings electrosynthesized on WBC alloy (**a**) and Hera type alloy (**b**) from an ionic-liquid electrolyte containing 0.5 M Py and 0.025 M Indo.

**Table 1 materials-15-04714-t001:** Chemical composition of the CoCr alloys.

Alloy Type	Composition/wt%									
	Co	Cr	Mo	Mn	Si	C	N	W	Fe	Ce
Wirobond C (WBC)	61	26	6	-	1	0.02	-	5	0.5	0.5
Hearenium CE (Hera)	63.5	27.8	6.6	0.6	1	0.3	0.2	-	-	-

**Table 2 materials-15-04714-t002:** Contact angles of CoCr based alloys, with and without PPy or PPy-Indo coatings.

Alloy Type	Coating	Contact Angle, Degree
WBC	without	86.7 ± 1.1
	PPy	27.4 ± 0.4
	PPy-Indo	20.7 ± 0.5
Hera	without	55.4 ± 0.9
	PPy	21.2 ± 0.5
	PPy-Indo	19.8 ± 0.4

**Table 3 materials-15-04714-t003:** The values of the equivalent electrical circuit elements for the uncoated and coated CoCr based alloys immersed in Tani-Zuchi artificial saliva.

Sample	R_s_, Ω·cm^2^	CPE_dl_ − T, µF·cm^−2^	CPE_dl_ − P	R_ct_, Ω·cm^2^	CPE_coat_, mF·cm^−2^	CPE_coat_ − P	R_coat_, Ω·cm^2^	Chi-Squared (χ^2^)
WBC	2.4	2600	0.842	7713	-	-	-	1.1 × 10^−2^
WBC-PPy	1.5	47.2	0.797	325	4.12	0.715	2467	1.7 × 10^−2^
WBC-PPy-Indo	2.29	27.39	0.79	701	1.56	0.507	7.15 × 10^8^	2.3 × 10^−2^
Hera	2.1	4269	0.809	3355	-	-	-	1.6 × 10^−3^
Hera-PPy	5.7	77.5	0.773	141	6.92	0.680	961	5 × 10^−3^
Hera-PPy-Indo	6.6	72.8	0.736	513	10.44	0.478	14,486	4.7 × 10^−3^

**Table 4 materials-15-04714-t004:** Kinetic corrosion parameters for uncoated and PPy and PPy-Indo coated CoCr-based alloys in Tani-Zuchi artificial saliva.

Sample	Tafel Method						Polarization Resistance Method	
	E_corr_, mV	i_corr_, µA × cm^−2^	Kg, G × m^−2^h^−1^	CR, Mm × year^−1^	Ba, mV	−Bc, mV	R_P_, Ω	i_corr_, µA × cm^−2^
WBC	−156 ± 0.03	78.82 ± 0.02	0.0976 ± 0.001	1.0052 ± 0.0007	83 ± 0.2	92 ± 0.3	277 ± 0.07	68.4 ± 0.04
WBC-PPy	96 ± 0.01	4.65 ± 0.01	0.0575 ± 0.0003	0.0593 ± 0.0003	79 ± 0.1	81 ± 0.1	5743 ± 0.1	3.02 ± 0.01
WBC-PPy-Indo	292 ± 0.02	0.55 ± 0.01	0.0068 ± 0.0001	0.0069 ± 0.0001	81 ± 0.2	80 ± 0.1	37318 ± 0.3	0.47 ± 0.005
Hera	−250 ± 0.03	181.2 ± 0.05	2.0125 ± 0.004	2.2028 ± 0.005	74 ± 0.2	89 ± 0.1	115 ± 0.07	152.6 ± 0.03
Hera-PPy	52 ± 0.01	17.7 ± 0.02	0.1968 ± 0.0007	0.2154 ± 0.0009	73 ± 0.1	93 ± 0.1	1067 ± 0.2	16.6 ± 0.02
Hera-PPy-Indo	181 ± 0.01	3.15 ± 0.01	0.0351 ± 0.0003	0.0383 ± 0.0002	84 ± 0.1	95 ± 0.1	6685 ± 0.2	2.89 ± 0.01

**Table 5 materials-15-04714-t005:** Parameters resulting from curve fitting using Equation (2).

CoCr Alloy Substrate	Parameters		
	n	k/h^−1^	r^2^
WBC	0.9564	7.293	0.9966
Hera	0.901	4.784	0.9896

## Data Availability

The data presented in this study are available upon request.

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
