# Peer review of "Simultaneously Embedding Indomethacin and Electrodeposition of Polypyrrole on Various CoCr Alloys from Ionic Liquids"

_materials, 2022, doi:10.3390/ma15134714_

Round 1
Reviewer 1 Report
very interesting and very current topic.
the introduction explains in an exhaustive way the topic of the work.
the materials and methods are very precise so that you can replicate the experiments without problem.
results explained in a simple and concise way and very simple and intuitive figures.
the conclusions supported by the results explained above.
just a few notes to point out:
all text should be checked for some spelling errors;
line 93: why 1200? why not more or less? explain briefly
figure 4: completely wrong scale and description!
figure 5: probably the legend and the colors of the spectra are reversed. Please check
Reviewer 2 Report
In this manuscript, Polypyrrole films were electrochemically deposited on commercial CoCr Wirobond C (WBC) and Heraenium CE (Hera) alloys using choline chloride-based ionic liquids at various potential, period times and simultaneously an indomethacin embedding and release. The content of the article is detailed and sufficient. I found the work proposed by the authors to be technically well-orgnized, as well as its characterization. However, before the article can be published, some questions need to be answered or some errors need to be corrected by the author.
1. What is the novelty in the current paper?. Please discuss in detail.
2. Readers would preferably want to know what you had concluded from all these studies, instead of what the author of the literature studies had concluded
3. Is it “choline chloride-malonic acid” eutectic solvent or ionic-liquid?
4. Check for the typo/grammatical errors throughout the manuscript to enhance the quality of manuscript.
5. Why did the authors select conducting polymer “polypyrrole” compare to “polyaniline, polycarbazole and polyindole” in this paper.
